# An Introduction to Nanopore Sequencing: Past, Present, and Future Considerations

**DOI:** 10.3390/mi14020459

**Published:** 2023-02-16

**Authors:** Morgan MacKenzie, Christos Argyropoulos

**Affiliations:** 1Department of Internal Medicine, Division of Nephrology, School of Medicine, University of New Mexico, Albuquerque, NM 87131, USA; 2Clinical & Translational Science Center, Department of Internal Medicine, Division of Nephrology, School of Medicine, University of New Mexico, Albuquerque, NM 87131, USA

**Keywords:** next-generation sequencing, nanopore sequencing, biosensors, single-molecule analysis, molecular diagnostics, genetics, transcriptomics, epigenetics

## Abstract

There has been significant progress made in the field of nanopore biosensor development and sequencing applications, which address previous limitations that restricted widespread nanopore use. These innovations, paired with the large-scale commercialization of biological nanopore sequencing by Oxford Nanopore Technologies, are making the platforms a mainstay in contemporary research laboratories. Equipped with the ability to provide long- and short read sequencing information, with quick turn-around times and simple sample preparation, nanopore sequencers are rapidly improving our understanding of unsolved genetic, transcriptomic, and epigenetic problems. However, there remain some key obstacles that have yet to be improved. In this review, we provide a general introduction to nanopore sequencing principles, discussing biological and solid-state nanopore developments, obstacles to single-base detection, and library preparation considerations. We present examples of important clinical applications to give perspective on the potential future of nanopore sequencing in the field of molecular diagnostics.

## 1. Introduction

Over the last three decades, methods for single-molecule detection—the third generation of sequencing methodologies—have broadened the scope of scientific research. Such methods make accessible cutting-edge sequencing technologies that enable a range of applications, from clinical discoveries to the characterization of protein kinetics [1]. The utilization of biological nanoscale pores to detect nucleic acid molecules promised the potential of making single-molecule sensing more accessible to a wider audience of researchers. A myriad of discoveries in the late 1990s surrounding nanopore use—including the theoretical conceptualization of using nanopores for nucleic acid sequencing [2]; the solving of the structure of staphylococcal alpha-hemolysin nanopore; the first biological pore used for nucleic acid translocation experiments [3]; and the proof of concept with alpha-hemolysin pore (αHL) [4]—in many ways marked the beginning of subsequent research in both biological and solid-state nanopores throughout the early-mid-2000s. Their large-scale application by Oxford Nanopore Technologies (ONT) in the 2010s [2] has made nanopore sequencing widely available, as these sequencers enable long-read sequencing and remain competitively priced compared to other platforms. The simplicity of these long-read sequencing systems makes these devices attractive for a range of applications, including genomic phenotype detection, structural variant detection, molecular biomarker discovery, and epigenetic research.

This review will provide a brief background on the development of nanopore sequencing technologies, beginning with an overview of the general construction and sequencing principles involved in nanopore devices. We will then discuss the multiple biological pore variants that have historically been used and those more recently developed, before delving into an overview of solid-state nanopore fabrication principles and materials. Using this background as a starting-off point, we will then discuss common applications utilizing nanopores, placing a specific focus on nucleic acid sequencing with high clinical relevancy. To introduce this discussion, we will open with library preparation technologies and recent modifications to these protocols, which permit sequencing-complicated sample types. The clinical applications of nanopore sequencing will include: an investigation into how innovative sequencing approaches have permitted viral genome assemblies; and the identification of rare genotypes, biomarkers, and epigenetic phenomena associated with disease. We will then conclude with a discussion of sequencing platform comparisons, providing insight to the differences between commercially available sequencing tools, such as Illumina, Pacific Biosciences (PacBio), and Oxford Nanopore Technologies (ONT). We hope this review will introduce nanopore sequencing for the curious reader, while also providing perspective on the promise of this tool for progressing the field of molecular medicine.

## 2. Nanopore Sequencing Principles

The idea of single-molecule detection on nanopore systems was independently conceptualized in the 1980s by several laboratories, including investigators David Deamer, George Church, and Hagan Bayley [5,6]. The original postulation was rooted in the theory that if exposed to an electrical current, oligomers could be driven through a protein nanopore channel, disrupting the current as they passed through, in a manner characteristic of their base composition. In principle, nanopore sequencing relies on a biological or synthetic nanoscopic pore spanning the length of a membrane that separate two chambers filled with electrolytic fluid (for example, KCl, or Ag/AgCl systems). The sequencing chamber lies on the cis side, while the chamber into which an analyte exists is termed the trans side [7,8]. Either chamber is connected to a voltage bias that distributes an ionic current throughout the nanopore, from the vestibule to the constriction site (protein structure [9] in Figure 1A) [6,8]. The mechanism is attached to a patch-clamp amplifier to permit the detection of the resultant signal (though this system has been compacted into portable ASIC chip systems by ONT [10]). In the case of nucleic acid analysis, the negative charge of the molecules causes them to drift away from the negative electrode, towards the anode, and through the nanopore. As they do so, each nucleic acid base interacts with the ionic current to cause a disruption in the current. These nucleotide fingerprints can be mapped back to both the length of the strand, generally, and the characteristics of its component bases, specifically [4]. The translocation of DNA or RNA through nanopores can be characterized by event duration (the time the molecule takes to move through the length of the pore), and the magnitude of the current blockade during translocation [1]. These quantities underline the conversion of the electrical signal into a readout appropriate for the sequencing application at hand. 

## 3. Biological Nanopores

### 3.1. Biological Nanopore Variants

The original experimentation of the detection of homopolymers and single-stranded nucleic acids by Kasianowicz et al. utilized *Staphylococcus aureus* alpha hemolysin (αHL), a secreted pore-forming toxin that inserts into the bilipid membrane of its host, causing osmotic disruption and cell-lysis (Figure 1B) [3,4]. The original αHL pore, characterized by Song et al. in their 1996 study, demonstrated that the pore was a heptamer consisting of a ~2.6 nm diameter transmembrane channel composed of 14 antiparallel beta strands, an area narrow enough to accommodate single-stranded DNA (ssDNA). This early experimentation verified the ability of the αHL nanopore to allow the passage of ssDNA, while also demonstrating that each ssDNA translocation disrupted the current in a measurable fashion [4,11]. Single-base recognition was later demonstrated using αHL [12], and the utilization of multiple recognition sites (sites within the pore where base recognition occurs) demonstrated a potential method to improve distinction between base identities (the “letters of the nucleic acid alphabet”) [13].

However, other biological nanopores with similar activities have been characterized and used for single-molecule detection studies. Aerolysin protein, a pore-forming toxin secreted by *Aeromonas hydrophila,* has a stem that is 1.0–1.7 nm in diameter, an ideal size for nucleic acid sequencing (Figure 1C) [14,15]. Further, Cao et al. successfully used aerolysin to detect variably sized deoxyadenosine chains and characterize the catalytic activity of endonuclease I [15]. This work notes that the small diameter of the aerolysin pore, in addition to electrostatic interactions between the pore wall and individual nucleotides, enable sensitive single-base pair discrimination. 

The outer membrane protein G of *Escherichia coli* has also been investigated as a potential nanopore for sequencing, given the fact that its constriction site measures around 1.3 nm (Figure 1D) [16,17]. However, this 33 kDa protein has open and closed conformation states that are pH and voltage dependent, and therefore it undergoes spontaneous gating events that makes it difficult to implement for single-molecule sensing [16]. Through molecular dynamics simulations, Chen et al. identified a key aspartate residue at position 215 in the protein that, when mutated, decreased overall gating events per second [17]. With a double mutant consisting of the D215 deletion, and an engineered disulfide bridge in beta chains 12 and 13, the Chen group was able to reduce gating events that might otherwise complicate single-molecule detection. With the addition of a cyclodextrin adapter within the pore’s barrel, this mutant is also able to detect ADP molecules. This pore has been more recently optimized through an alternative double mutation—the deletion of residues 221-227 and a mutation of the arginine at position 228—which also appear to decrease gating events [18].

A popular choice in nanopore sequencing use is a pore produced by *Mycobacterium smegmatis*, which has a smaller constriction site (~1.2 nm [19]) than the αHL pore, making it better for single-nucleotide resolution (Figure 1A) [20]. However, negative amino acid residues around the rim of the pore originally complicated analyte detection. Interestingly, Manrao et al. engineered a mutant of this pore, designed with a neutral instead of a negatively charged mouth [20]. Utilizing a NeutrAvidin anchor to immobilize ssDNA into the pore, MspA was found to: (a) sensitively detect residual current levels characteristic of immobilized homopolymers of each nucleotide type; (b) distinguish them based on their orientation (5′ or 3′ entry); and (c) detect characteristic current differences between methylated and unmethylated cytosines [20]. The same group found that the region of sensitivity for this pore was approximately 14.5 nucleotides away from its anchor, and using this information, reported the detection of single-nucleotide polymorphisms (SNPs) associated with breast or prostate cancer in genomic segments. The detection of the mutant MspA was improved with the addition of a molecular motor to the DNA that can dock onto the pore, a design motivated by the need to slow the translocation speeds of DNA below their natural rates (see below) [21]. More recently, the sensitivity of the MspA system has been used to characterize the kinetics of helicase enzymatic activity [22].

Recently, ONT introduced mutants of the curli transport lipoprotein, CsgG, to their nanopore devices [23]. This pore has reportedly been used in DNA sensing [24] and direct RNA sequencing applications [25]. While the constriction site of CsgG is narrow enough to permit sensitive base pair discrimination (~1.5 nm), a recently developed mutant has introduced a second constriction by inserting the naturally occurring accessory protein—CsgF [26]—to the interior of CsgG (Figure 1E) [27]. This mutant demonstrates improved single-base resolution during DNA sequencing on ONT platforms [27].

### 3.2. Slowing Translocation Speeds in Biological Nanopores

Modulating the translocation speed through motor protein facilitates the detection of electrical signatures associated with the passage of specific nucleotides through the pores. If the translocation speed is too fast, the current blockade will not be able to be detected without complicating high-frequency clocked electronics. On the other hand, slow translocation implies that the single-molecule process itself will take an extremely long time to conclude, thus compromising the use of nanopore devices for real-time sensing. In many *biological* nanopore systems, the bacteriophage phi29 DNA polymerase (phi29 DNAP) has been used as a molecular ratcheting system to slow the translocation of nucleic acids through the nanopores, via controlled 5′-3′ synthesis. Translocation through MspA was slowed by docking a phi29-DNAP-DNA complex to the nanopore, where the polymerase synthesizes the DNA complement into the pore in a controlled manner [21]. The pairing of MspA to phi29 DNAP was able to slow DNA translocation and enable single-base discrimination. However, more recently helicase enzymes have been employed for this purpose, as they have been found to produce more sensitive current alterations and slow translocation to approximately 450 bp/sec for DNA [28,29,30]. To this end, it should be noted that these molecular motors are ATP dependent, and therefore the continuity of the sequencing experiment will depend on a consistent fuel source. Currently, a fixed amount of fuel is loaded at the beginning of the experiment and its depletion over time eventually leads to the termination of sequencing. 

## 4. Solid-State Nanopores

While biological nanopores have been extensively developed and are a robust system, their relative shelf life, their limited reuse potential, and the difficulty in engineering them to exacting levels make them a less than ideal route of nanopore sequencing. Additionally, both the size of the constriction site of the nanopore trunk and the thickness of the membrane employed should ideally be close to the size of the analyte in question, to increase the sensitivity of detection; creating biological nanopores with this stringency is difficult [31]. Recent work with solid-state nanopores (SSNPs) in silicon-based and 2D atomic-sized membranes are promising solutions to this problem. They are broadly considered to hold a place in the future of nanopore sequencing as they can be fabricated with high precision, are robust against high voltages and other experimental parameters, and can be integrated within microfluidic devices [23,31,32].

### 4.1. Fabrication Techniques

In constructing SSNPs, drilling nanopores into the deposited material of choice should be precise and reproducible to facilitate accurate sequencing. Several techniques have been established, including, but certainly not limited to: focused ion beam (FIB) drilling/sculpting (often with Ar^+^ [33] or Ga^+^ [34] ions); the transmission electron microscopy (TEM) drilling/sculpting [35] laser pulling of glass pipettes to create glass capillaries [36,37]; evaporation induced self-assembly [38]; and controlled dielectric breakdown [39,40]. (For thorough reviews on the creation of SSNP fabrication methods, we encourage readers to refer to the following reviews [31,41,42,43]).

Ion-beam sculpting was originally tested for SSNP construction [24]. Li and collaborators demonstrated in 2001 that by exposing a Si_3_N_4_ membrane with concave indentations to 3-KeV Ar^+^ ions, atoms can be stripped from the surface of the silicon-based membrane in a feedback-controlled manner, thinning the membrane and eventually forming a pore with the indentations on the opposite side [33]. This technique was utilized to create a ~5 nm diameter pore capable of detecting dsDNA with current reductions of up to 88% of the pore’s center. Of note, this work demonstrated that under excessive ion-beam exposure, lateral atomic flow redeposits the material across the nanopore opening, effectively closing it. Using this feedback strategy, subsequent work drilled 100 nm diameter nanopores with FIB in silicon nitride membranes, subsequently shrunk to diameters near 3 nm [44]. This small size (for reference, just slightly larger than the diameter of αHL) permitted the analysis of various levels of dsDNA folding and intermolecular pairing [44]. FIB drilling has further been paired with ion scanning to create and modulate the size of an array of nanopores below 20 nm, and down to 5 nm [33].

Transmission electron microscopy (TEM) has also been employed as an electron beam sculpting system to permit the real-time analysis of the pore size and the direct sculpting of the pores [35,41,45]. Storm’s group utilized electron beam lithography to create pores within a silicon oxide membrane, which were then shaped to variable dimensions with TEM [46]. The group found that the mechanism of pore size alteration depended on the starting material’s thickness, where pores >80 nm could be widened and those under 40 nm would shrink. This controllable shrinking strategy was later determined to be the result of surface-tension induced mass flow, resulting from the fluidization of the SiO_2_ material induced by TEM [47]. (For a thorough review of controllable shrinking strategies in SSNP systems, we direct the reader to other reviews [43,48]). 

While TEM and FIB are powerful techniques for SSNP construction, they are expensive and require specialized equipment that may not be available to every laboratory. Additionally, while nanopores can be crafted before implementation in a fluidic chamber, the introduction of prefabricated nanopores into electrolyte solutions may alter the characteristics of the pores of certain materials [49]. In situ fabrication methods, such as controlled dielectric breakdown, alleviate this concern, while also being more accessible and inexpensive methods for a wider range of laboratories [42]. Using the work of Kowk et al. as an example, controlled dielectric breakdown permits the formation of nanopores in solution by distributing a potential difference across the surface of a dielectric membrane, creating a strong electric field that creates nanopores in the surface of the starting material (in their case, silicon nitride) as a consequence of the induced charge build up [40]. While these pore sizes can be fabricated to ~1 nm diameters, it is difficult to control their location on the surface of the membrane, which may result in irregularly sized pores [31,39]. Notably, controlled breakdown has been applied to silicon nitride membranes embedded in microfluidic devices, permitting the enhanced detection of dsDNA and proteins [49].

### 4.2. Materials for Construction

Although the list of explored SSNP materials is extensive, a few of the commonly tested materials of interest are silicon-based, glass capillaries, graphene monolayer assemblies, and molybdenum disulfide (MoS_2_) layers [31,50]. The choice of the material largely influences the maximum voltage that can be used during translocation experiments (as certain materials are much more robust than others, which will start to erode under high-voltage stressors [51]), as well as what chemical modifications can be added to the surface for enhanced analyte sensing (see below) [52]. Beyond this, constructing membranes of significant thinness is important for maximizing single-base resolution, as it limits the number of bases that contribute to the current disruption (Figure 2) [53].

Silicon-based systems have been proposed as a material for this application; because their theoretical thickness can be narrowed down to extremely thin dimensions, they can withstand high-voltage biases, and can be operated under greater bandwidths with decent resolution [51,54]. Rodríguez-Manzo et al. developed an electron irradiation-based technique for silicon membrane nanopore preparation, with the goal of maximizing the conductance detection of the nanopore while decreasing baseline signal noise [55]. They scanned silicon nitride films with scanning TEM to decrease the thickness of the silicon membrane, then used an electron probe to bore the nanopores into the surface of the synthetic membrane. The thinnest membrane achieved in this work was 1.4 +/− 0.1 nm, and each pore had a diameter of 1.3–2.4 nm (comparable to the diameter of the αHL pores). Of note for silicon membrane construction, molecular dynamic simulations have demonstrated that the physical bottle neck for Si nanopore membranes is around 0.7 nm, after which point they are no longer stable [55]. Silicon-based nanopores have also been constructed on glass chips to enhance stability and reduce the capacitance of the silicon membrane, allowing for translocation events to be detected with short-event durations [56].

Graphene, a monoatomic layer of carbon grid structures, has many attractive physical qualities including electrical conductivity, malleability, and its impenetrability to ions and experimental parameters, e.g., pH and temperature [57]. Unlike silicon, graphene is stable as a monolayer system, and can exist comfortably at a thicknesses of 0.3 nm [53]. Techniques have successfully grown graphene membranes on silicon chips and used electron beam drilling to create nanopores around ~3.3 nm in size, enabling measurable differences between ssDNA and dsDNA [58]. However, similar 2D materials have also been proposed for use as nanopore systems. MXene membranes—atom-thick layers of transition metal carbides [59]—have also been proposed as a potential substitute for graphene. Specifically, Ti_3_C_2_(OH)_2_ nanopore membranes have been studied in the theoretical evaluations of sequencing efficiency and were found to allow for base distinction between all four bases in molecular dynamics simulations [59].

### 4.3. Controlling Noise and Translocation Speeds in Solid-State Nanopores

The implementation of SSNPs in nucleic acid sequencing is currently limited by the difficulty associated with slowing translocation speed, which is needed to enhance the signal-to-noise ratio (SNR) of analyte translocation. Previous investigation has proposed that background, low frequency (1/*f*) noise interferes with SNR in both biological and solid-state nanopore systems, likely due *in part* to: (a) nanoscopic gaseous bubbles within the pore (nanobubbles) [60], or irregularities in the pore structure in the case of SSNPs; (b) conformational changes in the case of biological pores; and/or (c) electrode noise in either case [51]. While other methods of enhancing the signal-to-noise ratio of solid-state pores exist, an importance is placed on slowing down translocation time, enhancing the signal, and minimizing these background frequencies. DNA naturally threads through nanopores at a rate of ~ 1 million bases/second, which is too fast for single-base resolution with current electronics and computational techniques for signal processing. There are techniques which can be employed to slow nucleic acid translocation, including molecular motors and mutations within biological pores (as discussed above), and chemical group additions to the surfaces of synthetic nanopores and their membranes [52,61].

Such chemical modifications have been extensively studied and include: chemical and physical vapor deposition; atomic layer deposition (ALD); chemical group modifications of the surface layer; the coating of solid-state surfaces with lipid bilayers; and the creation of hybrid nanopores by inserting biological pores with solid-state tunneling [31,62,63,64]. By altering the charges of the surface and/or adding chemical components or probes for specific analytes, the interactions between the sequencing nucleic acid strands and the SSNPs are thought to be enhanced, improving the likelihood of successful and trackable translocation. For example, Wang and colleagues recently developed Gold-Fe_3_O_4_ nanoparticles, adapting them with peptide nucleic acids that can bind to targeted short-RNA molecules [65]. These complexes enabled the detection of RNA translocation through glass quartz nanopores. The range of these applications is broad, and the methodology for chemical functionalization can be chosen by the investigator to produce desired interactions with the analyte of choice.

## 5. Oxford Nanopore Technologies vs. Other NGS Platforms

As aforementioned, the commercialization of nanopore sequencing by Oxford Nanopore Technologies (ONT) in 2014 has made protein nanopores widely available for use in many research applications. They currently dominate the nanopore sequencing space, providing sequencers and library preparation kits for DNA and RNA nucleic acid sequencing. As the popularity of these platforms grows, we recognize that many researchers may be seeking comparative information between ONT and other available options. While a thorough comparison of these platforms is beyond the scope of this review, we wish to provide a brief perspective on this issue, focusing on differences between Illumina, PacBio, and Oxford Nanopore Technologies.

### 5.1. Short vs. Long Reads

At the time of writing, there are a handful of popularly used sequencers that fall under the classification of either short read (often referred to as second generation) or long read (third generation) platforms [66,67]. These platforms include, among others, Illumina and Ion Torrent sequencers, which are the short read platforms largely used in current research. To describe an average Illumina sequencing workflow in brief: nucleic acid libraries of long lengths are fragmented (more so for DNA, not always for RNA), producing fragments of lengths between 150–800 bases long [67]. The sequences are then put through an end repair step to correct the damaged strands and prepare them for the attachment of sequencing adapters necessary for attaching the fragments to the surface of the Illumina flow cells. Following adapter attachment, a size selection is then performed to produce proper fragment sizes, and the sequencer is loaded [68,69]. 

Both Illumina and Ion Torrent sequence nucleic acids by means of “sequencing by synthesis” (SBS) [68,69]: fragments are attached to scaffolds along a flow cell surface, and a DNA polymerase enzyme synthesizes the complement of each strand. The principle of Illumina sequencing rests on the incorporation of fluorescently labeled bases that pair with the DNA complement. Each of the four bases is given a characteristic color so that upon binding, the color can be detected via real-time image analysis and traced back to its base. The base is then removed, allowing space for continued synthesis, and the reading of bases further down the strand. Both DNA and RNA can be sequenced on Illumina sequencers, though RNA-sequencing takes place by means of cDNA synthesis [69]. 

Following the establishment of short read platforms, third generation sequencers, like the nanopore platforms that are the focus of this review, were explored as a means to navigate through the limitations inherent to short read sequencing. Because second generation sequencers require short fragments of nucleic acid to be compatible with the flow cell surface, postsequencing these short reads must be reassembled, requiring complex bioinformatic pipelines that may not be widely user-friendly [67]. Additionally, the fragmentation of reads complicates the processing of repetitive regions and low sequence diversity, such as ribosomal DNA repeat sequences that are linked to disease, short tandem repeat sequences, sequence isoforms, and structural variants [67,68,70]. Of note, short read platforms do not enable direct RNA and DNA sequencing, limiting the ability to explore epigenetic modifications relevant to disease processes (discussed below).

### 5.2. ONT vs. PacBio

The two most widely used long read, third generation sequencers in today’s laboratories are ONT and Pacific Biosciences sequencers (PacBio sequencing, or single-molecule real-time (SMRT) sequencing). PacBio sequencing technology relies on adding a double stranded DNA/cDNA molecule with hairpin adapters to the 5′ and 3′ ends [70]. The libraries are inserted onto a sequencing chip that house an array of reaction chambers, each with a DNA polymerase set at the bottom of the well. The DNA polymerase then attaches to the hairpin primers and replicates the DNA sequence. Like Illumina sequencing, PacBio relies on SBS and photometrics, utilizing fluorescently labeled dNTPs to identify the bases. Upon incorporation, each nucleotide releases a flash of light utilized for base detection. The fluorescent dye is then dissociated from the base and the cycle continues, producing light patterns that can be base-called to the original sequence identity [70]. As discussed in detail above, instead of photometrics, ONT employs the electrical detection of base pairs. In this way, the technology is rather novel from previously established SBS methods and provides the potential for direct nucleic acid sequencing analysis.

Historically, long read sequencing platforms have suffered from high error rates associated with base detection, where rates in the 2010s fell between 11–13% for PacBio and around 38% for ONT sequencing [70]. However, these error rates have improved dramatically with changes in sequencing methodology, the software for base detection, and additional library preparation modifications. 

For PacBio, these improvements include consensus sequencing (continuous long read (CLR) or high fidelity (HiFi) sequencing): this utilizes the DNA polymerase in each sequencing well to repeatedly sequence the same read, incorporating each pass into a final consensus sequence [68] (though notably, the limited half-life of the DNA polymerase makes this method mostly suitable for reads around 20 kbp [71]). This compensates for sequencing errors associated with individual subreads, and brings the accuracy of new PacBio sequencers up to >99.8% [68]. Oxford Nanopore Technologies also experimented with this concept early on, though in the nanopore realm, this manifests as sequencing both reads of a single- (C)DNA-molecule that is either adapted with a hairpin adapter (2D sequencing) or that has one strand tethered to the flow cell surface, making tandem sequencing easier (1D^2^); both have been shown to produce accuracy rates upwards of 99.5% [72,73]. However, the incorporation of unique molecular identifiers (UMIs)—barcodes that contain a randomized sequence for each read, permitting bias-compensation for library preparation steps—into the library prep for PacBio and ONT has been shown to decrease these error rates even further [74]. This is becoming a viable option for users to explore error-rate correction. An alternative option for producing highly accurate reads involves hybrid-assembly, which incorporates long and short read platform data to piece together reads, compensating for the shortcomings in either technique [75]. 

While both of these platforms are used to produce long read sequences, the high accuracy HiFi approach on PacBio produces reads with N50 (that is, the length of the shortest read within the group of the longest reads that constitute at least 50% of the sample) of 10–60 kb [76,77,78], while ultralong reads on ONT sequencers have been shown to produce upwards of three to four megabases (Mb), with N50 of 100–200 kb being possible (though 10–30 kb reads are common for everyday long read DNA sequencing) [75,78,79]. 

Previous cost-comparisons between PacBio and ONT demonstrate significant differences in cost, depicted in a recent review [78]. For PacBio’s Sequel II platform, generating maximum read lengths >200 kb with a 87–92% read accuracy platform, the estimated cost per gigabase (Gb) was USD 43–86 /Gb; the comparable ONT platform, the PromethION, producing maximum reads >1000 kb with a 87–98% read accuracy, has an estimated cost of USD 21–42 /Gb. 

An additional consideration of comparison between these sequencers is application. While both platforms are able to sequence DNA and RNA inputs, only ONT offers direct RNA sequencing. Finally, for users interested in portability for field sequencing, ONT MinION sequencers are noted for their small size; a plus for those interested in mobile, in the field sequencing. This can be a useful characteristic for applications requiring mobile sequencing technology and is currently lacking for PacBio or Illumina sequencers. 

## 6. Library Preparation Considerations for Nucleic Acid Sequencing

### 6.1. Library Preparation Overview

Having been one of the preliminary motivations for the 1990′s exploration of nanopore platforms, nucleic acid sequencing is perhaps the most common application of nanopores. Both direct and PCR-amplification methods have been used to execute DNA and RNA sequencing on nanopore platforms [80]. To facilitate sequencing, samples must be converted into the proper format for the sequencing platform in question, through a process called library preparation. Library preparation effectively functions as both preanalytic signal filtering (excluding molecules that are of no interest to the user) and as the signal amplification of low-input samples through nucleic acid amplification techniques, such as polymerase chain reaction and rolling circle amplification. Library construction methods for DNA samples can easily be modified and adapted to the user’s sample to permit the highest possible coverage of an analyte, e.g., genome or transcriptome, or to focus attention to a particular molecule (or groups of molecules), i.e., targeted sequencing. The library preparation principle is similar across nanopore and non-nanopore based sequencing platforms. We review the salient details of the process for Oxford Nanopore Technologies [81]. Further information on library preparation details can be found in more expanded reviews [30,82].

For many long read DNA samples, the first steps involve the fragmentation of long sequences of DNA into smaller units, followed by the end-repair enzymatic reactions of the damaged DNA [30,81]. This ensures a degree of uniformity of the molecules that pass through the pores. This mechanism often involves the adenylation of DNA ends, making them compatible to hybridize with single-thymine overhangs attached to *sequencing adapters*. The adapters are subsequently attached via enzymatic covalent ligation, or rapid-attachment chemistries [30]. These sequencing adapters make the DNA library compatible with the nanopore platform, but also serve as: a) reverse transcription and strand switching primers in the case of RNA → cDNA conversion protocols; and/or b) PCR primers in the case of amplification techniques. These primers can simultaneously serve as molecular barcodes for multiplexing experiments, which have the potential to lower the cost of sequencing significantly by enabling the sequencing of multiple samples at a time. DNA can also be sequenced natively (without amplification) to avoid the introduction of PCR bias into the sample. However, in the case of limited DNA input (<100 ng [81]), amplification is useful to provide acceptable library depth for whole genome or targeted sequencing applications. 

Following the attachment of sequencing adapters and the optional reverse transcription and/or PCR amplification, side reaction products, e.g., primer dimers, can be enzymatically degraded with a DNase enzyme (an important step to maximize the library depth of target sequences). Further, libraries are cleaned using either bead-based or column-based methods, both of which can be adapted to the sample content for sufficient retention. Finally, a second sequencing adapter is added to both 3′ and 5′ ends of the polymers just before initiating a sequencing experiment. This adapter includes a helicase motor that attaches to the nanopore and helps ‘unzip’ double-stranded polymers, translocating them as they enter the pore. This step is also crucial to fine tuning the resolution of current disruptions, as it slows the translocation of the DNA molecules down to approximately 450 bp/s [30], a speed that allows the sequencing of DNA molecules and an analysis of the electrical signals with simple electronics.

In the case of RNA library preparation, the layout is similar: cDNA libraries can be created from RNA samples using reverse transcription; or RNA can be sequenced directly [80,81]. In the latter case, an optional single-stranded cDNA synthesis can be performed to limit the formation of complex RNA secondary and tertiary structures (though this strand is not sequenced through the nanopores). Prior to sequencing, sequencing/motor protein adapters are attached through enzymatic ligation to the 3′ end of the RNA product (direct sequencing) or rapid attachment to both strands (PCR-cDNA sequencing). While direct RNA (and DNA) sequencing can be utilized to preserve epigenetic modifications during sequencing (as in [25]), large amounts of RNA (>500 ng) must be input to account for inevitable losses [80,81]. RNA is also sequenced at slower rates (~70 bp/second), so the overall yield from these experiments is lower than DNA runs [30].

### 6.2. Library Preparation Modifications for Diverse Samples

Optimizations to library preparation methods have been executed to permit the sequencing of diverse libraries. Often, nucleic acid enrichment strategies are necessary to increase the library depth of targets. For example, running samples through ribosomal RNA or transfer RNA depletion prior to library preparation can limit the input of undesirable species to the library prep [83,84]. The Cas9-mediated sequence-specific adapter addition has also been proposed to select targets for downstream analysis [85]. A modification of this Cas9 protocol has been utilized alongside custom bioinformatics pipelines to detect fusion-pairs and breakpoint locations in cancer cell lines [86].

Further optimizations can be employed before or during library preparation to permit the sequencing of otherwise ignored species. For example, “Phospho-seq” utilizes a T4 polynucleotide kinase enzyme to add 5′ phosphates and 3′ hydroxyl groups to the ends of rarer subspecies of RNA that lack 5′ phosphates or contain 3′ phosphate groups [87]. This makes them compatible for the adapter attachments that are necessary during library preparation, and may also improve the polyadenylation-base enrichment methods [88]. Recent work by our group has demonstrated the ability to perform highly effective cosequencing of short and long coding and noncoding RNA species through universal poly-adenylation tailing, enabling the contextualized quantification of all RNA species [89]. To further enhance the detection of target species, spike-in synthetic nucleotide mixes permit quantification, which we have demonstrated with mixed RNA samples using ERCC spike-in mixes [89], and others have demonstrated with RNA isoform identification and quantification using synthetic sequin RNAs [90]. 

## 7. Clinical Applications: Nucleic Acid Sequencing on Nanopores

While nanopores have been applied to numerous areas of focus, including water purification [91], protein identification and characterization [32,92], and recently data storage [93], we will focus on nucleic acid sequencing for the applications section of our review. Other reviews explore the wide field of nanopore technologies beyond this scope [94]. The applications mentioned refer to nanopore sequencing on ONT platforms, unless otherwise stated.

### 7.1. Genomics and Structural Variants

#### 7.1.1. Genome Assembly

In 2018, Jain et al. developed an ultralong read method to successfully sequence a reference human genome de novo from the GM12878 cell line using Oxford nanopore’s CsgG mutant R.9.4.1 [79]. Their method included native DNA sequencing for the sake of accurately detecting repetitive elements and epigenetic modifications. This enabled them to achieve 30x genome coverage, 99.88% sequence accuracy, and high levels of agreement with competitive short and long read platforms. The group also successfully profiled complete MHC locus (an application directly applicable for the optimal matching of human organs between donor and recipient in clinical transplantation), estimated telomere lengths, and methylation profiles [79]. This accomplishment demonstrates the adaptability of the nanopore system to optimize the platform for ultralong reads (>100 kb), while helping to fill in the holes of reference genomes in hard to reach regions currently unattainable by short read sequencing platforms.

#### 7.1.2. Fusion Gene Detection

Fusion genes have also been a targeted application of nanopore sequencing, as fusion events are responsible for several forms of cancer. Identifying fused genes quickly in clinical samples is important to influence rapid treatment responses. The gold standard of fusion gene identification is fluorescence in situ hybridization, but the technique has a turnaround time of up to 48 h and may be insensitive to some mutations [95,96]. Nanopore sequencing has been applied to these sample types because of its potential to deliver fusion gene readouts rapidly within 12 h. Using a DNA adapter-ligation sequencing approach combined with modified bioinformatics, Jeck et al. were able to successfully identify the BCR-ABL1 gene rearrangement (a diagnostic hallmark for chronic myeloid leukemia [96]) and the PML-RAMA fusion within seconds of sequencing, even with low library depth of the target fusions. While a common complaint of nanopore sequencing involves its high error rate, this group found that even low-quality base calls were mappable to the regions of interest, something that our group has also found [89]. 

Using similar methodology, the same group was later able to sequence these same libraries on ONT’s Flongle device—their smallest, single-use flow cell that generates up to ~2.8 Gb of data (with yields commonly falling near 1 Gb [97] with a cost below USD 100—and were able to capture all of the previously identified fusion genes and the fusion *CIC-DUX4*, which is embedded in a locus with a high number of repeats [98]. This is a promising finding: with improvements to the device structure and pore design, the inexpensive Flongle flow cell may prove to be an accessible diagnostic tool for both genomic and transcriptomic applications.

#### 7.1.3. Short Tandem Repeat Detection

Among the sequencing problems that have been historically difficult to solve are short tandem repeat (STR) sequences, as they contain repetitive sequences difficult for short read sequencers to localize and properly identify. The use of nanopores for STR sequencing remains a challenging prospect, as they tend to generate reads with a high proportion of errors that, among other considerations, increase with the length of the repeat [99], and are unpredictable, varying with the location of the repeat [100]. However, because the electrical squiggle signals from nanopore sensors elucidate characteristics of base identity regardless of the composition of the nucleic acid, this issue is likely more so a result of the *basecallers* used during sequencing to convert electrical signals to nucleic acid bases, than to the platform’s capabilities itself. This implies that algorithmic developments in the absence of any radical platform innovations may expand the application of nanopore sequencing in this space. 

In that regard, recent work compared Guppy to Bonito—two frequently used basecallers that utilize recurrent neural networks and convolutional neural networks, respectively [101]—to genotype autosomal and nonautosomal STR loci, and the SNPs within them. While the investigators were able to genotype most STR loci correctly, the basecalling was easily obfuscated by the presence of homopolymers near the STR, highly repetitively expressed elements, and high sequence similarity; in other words, the success of the basecaller was dependent on the sequence. In support of this, previous work has demonstrated the successful sequencing of the mitochondrial genome of *Schistosome haematobium* with the Guppy basecaller, notably sequencing a tandem repeat region 18.5 kb long [102]. Thus, the success of this commonly used basecaller may be dependent on the applied scenario (discussed further below). Methods have been developed to improve the error rates of basecalling. For example, a recent development translates each electronic signal from an STR unit into a matrix that is then converted to chromatic channels [103]. Deep convolutional networks are then used to infer the identity of each signal and allocate them to their designated STR regions.

#### 7.1.4. DNA Nicks

While the above-mentioned applications have been performed on a commercialized protein pore system, there have been investigations into the application of nucleic acid sequencing through solid-state nanopores. Recently, Athreya and colleagues used molecular dynamic simulations and electronic transport models to model the detection of single-strand breaks in DNA molecules in graphene and MoS_2_ pores [104]. It was found that in graphene, DNA nicks cause the increased dwell times of a few nanoseconds, which they attribute to hydrophobic interactions between the unphosphorylated DNA backbone point and the hydrophobic nature of the graphene. They found that, depending on the nucleotide characteristics of the strand break, DNA strands will denature within the pores at different voltage biases. Therefore, they propose that the nick intersection location and the nature of the surrounding bases can be identified by finding the characteristic voltage that causes strand dislocation. Interestingly, this effect was not found in the MoS_2_ membrane and was in fact the opposite. The DNA nicks cause the DNA to translocate to one side of the pore, leaving more room for the translocation of ions through the pore, thus decreasing the detected electronic signal while increasing the current passage. Applying these principles to a 2D material sequencing experiment is a task for future SSNP studies.

### 7.2. Epigenetic Modifications

Epigenetic modifications of genomic material is likewise an attractive field for discovering novel mechanisms of disease development and the control of gene expression [105,106]. As an example, the methylation of DNA is a well-studied mechanism of transcriptional silencing, however, historically the transient nature of methylation and the low sequence complexity of highly methylated regions have complicated sequencing methylated regions [107]. The gold standard for methylation studies is the bisulfite conversion technique, which converts methylated cytosines (5meC) into uracil residues; however, this approach may risk confounding these converted methylation patterns with experimental error [108]. Methylation studies have successfully been performed on nanopore instruments. Davenport et al. utilized standardized nanopore sequencing kits with the older R9.5 chemistry to identify the hypermethylation of cytosines, enabling the discovery of potential tumor suppressor genes that may be epigenetically silenced in hepatocellular carcinomas [109]. The nanopore identification of genome-wide 5meC had high levels of agreement with standard bisulfite conversion and managed to discover 482 methylated genes that were invisible to short read sequencing platforms. However, the accuracy of these methods needs further verification and improvements to bioinformatic pipelines. In particular, for users interested in modification analysis, special attention should be paid to the basecaller employed, as this will largely determine the accuracy of read identification (as we discuss below). For methylation analysis, previous assessments have outlined user-specific suggestions, noting that the Guppy and Nanopolish software are sufficient tools for laboratories with limited computational hardware [110]. Additional methods have been developed to improve the detection of methylated residues from the ionic current signal data of ONT nanopore sequencers using hidden Markov models combined with hierarchical Dirichlet processes [111].

### 7.3. Infectious Disease Detection

The rapid detection of viruses and other pathogens is important to mitigate outbreaks and improve treatment in clinical environments. Current methodology is slower and more expensive than optimal, while targeting specific species may inhibit the detection of low-concentration targets and miss important species [112]. A platform that offers improvements to both techniques could dramatically improve disease control. Utilizing long read sequencing permits the detection of full-length pathogen genomes, while also enabling the characterization and identification of variants. Of note, ONT MinION sequencers have been employed to track numerous outbreaks, including the Ebola virus [113] and influenza in 2015 [114] and the Zika virus in 2016 [115]. Notably, MinION sequencers were recently used to track the SARS-CoV-2 Alpha and Delta variants in Ukraine, utilizing reverse transcription-driven cDNA sequencing [116]. Similarly, a group successfully sequenced a full monkeypox viral genome—including ITR sequences previously missed by other short read platforms—within 8 h, obtaining sequencing depths of ~12–57x genome coverage [117]. Although the platform struggled with detecting homopolymers greater than a few base pairs long, which is still an issue of the *basecaller* more than the basic sequencing principles, the authors note that they had success utilizing HomoPolish [118] to clean and correct some mismatches in their readouts. While these innovations are very promising, the library preparation and data analysis pipelines of these systems will need to be improved and normalized before this becomes a standard diagnostic tool. 

### 7.4. RNA Sequencing

Previously established RNA sequencing methodologies—those that rely on cDNA synthesis and short read production—are limiting in that they introduce bias and complicate the identification of full-length transcripts, isoforms, and modification patterns. While ONT platforms support cDNA-based RNA sequencing, one of the unique aspects that marks it apart from competitor platforms is its ability to sequence RNA directly (direct RNA sequencing), permitting the analysis of RNA modifications and limiting bias introduced by library preparation steps involved in cDNA synthesis. Similar to some approaches for DNA sequencing, long read products from nanopore have been combined with short read products to produce full-length transcripts with higher accuracies (though de novo transcriptomic assemblies have been created using cDNA synthesis methods [119]).Improvements to library preparation approaches, as discussed above, have permitted the sequencing of numerous RNA species on nanopore platforms, ranging from microRNAs [65,120] to tRNAs [121,122], and circular RNAsg [123], as well as the full-length transcriptomic isoform identification and determination of alternative splice sites [124,125].

#### 7.4.1. Full-Length Transcript Assembly

In 2019, Workman et al. successfully assembled a human B lymphocyte poly(A) transcriptome by using this technique, which identified several splice-junctions and novel isoforms, allele-specific isoforms, N6-methyladenosine modifications, and Adenine-to-Inosine editing [124]. Their direct RNA sequencing produced 50,000–831,000 reads per sequencing run (note that direct RNA sequencing will yield lower reads than cDNA approaches), with passed reads of N50 lengths up to 1334 bases. Of note, this work characterized the 5′ truncation patterns of direct RNA sequencing. This phenomena is likely attributable to errors during sequencing (helicase enzyme separation or stalls during transclocation, or strand breaks during the sequencing) [124,126], but recent developments have since been proposed to address this problem [127]. 

Along the lines of viral transcriptomics, Depledge et al. employed direct RNA sequencing to study the HSV-I transcriptome, employing a novel method to correct erroneous base calls [125]. Their method used Proovread [128] to align nanopore reads to a previously sequenced Illumina sequence, and from those corrections generated pseudotranscripts to identify read identities. While this approach resulted in improved mapping rates, the authors note the limited applicability of the technique outside transcript isoform studies [125]. 

#### 7.4.2. MicroRNA Detection and Quantification

MicroRNAs are short (~22 nt) RNAs that have important roles in gene expression regulation and disease development [129,130,131]. MicroRNA sequencing on nanopore platforms had been explored earlier [132], though there seems to be limited application on the commercialized ONT platforms. The recent investigation of microRNA sequencing coupled a MspA porin to phi-29 DNAP ratcheting protein and created chimeric microRNA-DNA hybrids for sequencing [120]. Their method permitted the discrimination of isoform and methylation markers, though did not capture the full microRNA body. Interestingly this work suggests that the microRNA sequencing on ONT sequencers may be complicated by their short length, though subsequent investigation by our group has demonstrated that with library preparation modifications, the platform is capable of detecting and quantifying microRNA sequences with bias on par with Illumina [89]. A different approach hybridized microRNAs to hairpin DNA sequences, and determined unzipping patterns of the duplexes to establish signals characteristic of microRNAs involved in bile-duct cancer [133]. Developing a protocol for direct microRNA sequencing would be a valuable step in detecting microRNA expression patterns, tailing motifs, and chemical modification patterns that have been connected to disease development.

#### 7.4.3. CircRNAs

Circular RNAs (circRNAs) are other noncanonical RNA species that have been demonstrated to regulate gene expression through interactions with other noncoding RNAs [134]. CircRNA sequencing has also been performed on ONT sequencers, with protocols utilizing rolling-circle reverse transcription to capture full-length transcripts, enabling the detection of isoforms and fusion reads, as well as alternative splice sites [123,135].

#### 7.4.4. Single Cell RNA Sequencing

Long read sequencing platforms offer exciting potentials in the realm of single cell RNA dynamics by permitting the improved exploration of alternative splice variants, post-transcriptional regulation, and RNA diversity between specific cell populations. A protocol developed by Lebrigand et al. in 2020 spearheaded the single cell sequencing applications of ONT. Their work employed UMI’s to compensate for experimental bias and errors, enabling the discovery of numerous novel and cell-type specific transcript isoforms with accuracy rates above 99% [136]. This and similar work [137] on nanopore devices will continue to accelerate the field of single cell transcriptomics in the near future.

## 8. Conclusions

Within the past decade, there have been several improvements in the field of nanopore sequencing, making it a competitive technology for fundamental science research and clinical diagnostics. While innovations to protein pores have been commercially used to this date and offer high single-base resolution, there is motivation for further development of solid-state nanopores, as they offer highly customizable platforms that are more robust than protein pores. Additionally, the potential of perfecting 2D materials such as graphene monolayers for nanopore fabrication would enable higher single-base pair resolution. Further research is needed to optimize these platforms, focusing on providing reproducible nanopore arrays by simpler, more accessible means. An important obstacle in this field is slowing the translocation speed of the analyte, which can be explored through the chemical functionalization of the membrane surface.

A significant body of work has gone into improving and modifying library preparation techniques for diverse sample types. With proper biochemical innovations, sample types that would otherwise go “unseen” by the sequencing platform can be captured and quantified. This is an important step in enabling the contextualization of disease processes, a crucial point for correctly analyzing epigenetic mechanisms and gene expression control. Additionally, the accuracy of nanopore sequencing can still be improved through changes in stranded sequencing. With ONT sequencers, the common “1D” sequencing approach does not currently permit sequential sequencing of both strands of a (c)DNA helix [31]. Previous attempts to permit this utilized a hairpin adapter that would anchor the other strand in place, waiting for the complete translocation of one strand before threading through the sister strand [138]. This approach enables an internal checking system for the basecaller, as it helps to verify the primary sequence with the information garnered from the complement. While it was phased out from the company’s platform for a few years, in 2022 this duplex sequencing seems to have been reintroduced, with some preliminary results suggesting improved sequencing accuracy [138].

Along the lines of improving read accuracy, one of the largest obstacles to large-scale nanopore sequencing implementation is the accuracy of basecalling. The variable basecalling accuracy noted in applications reflects the features of the neural networks based basecallers, which are supervised learning nonlinear models trained on predefined datasets. As with any neural net application, their reliability in recognizing bases (“feature detection”) is only as good as the original data they are trained on. Thus, for basecallers trained on data from a distantly related species that have different modification, e.g., 5-(hydroxy)methylcytosine, N6-methyladenosine, or repeat (e.g., homopolymers) patterns other than the user’s application, basecalling may not be as accurate as one would desire. 

Previous work comparing the performance of an array of basecallers relying on neural nets—Albacore, Guppy, Scrappie, Flappie (https://github.com/nanoporetech/flappie, accessed on 15 February 2023) and Chiron [139] —demonstrated that the performance (measured in terms of read accuracy (important for low read-depth samples) and consensus accuracy (a concern for high read-depth samples)) of Guppy can be improved by training the basecaller on large, taxon-specific datasets, as this produces learning of both base pair characteristics and modification contexts [140]. Until 2022, ONT provided Sloika (https://github.com/nanoporetech/sloika, accessed on 15 February 2023) for customized network training, but this has since been replaced by Bonito’s training capabilities (https://github.com/nanoporetech/bonito, accessed on 15 February 2023). For users sequencing heavily modified samples, utilizing a custom-trained neural network will likely lower error rates and improve single-base detection accuracy. However, for those with limited computational resources, there are numerous pretrained basecallers that have been evaluated for performance in different scenarios. Additionally, other tools exist to improve sequence-specific basecalling that help customize the analysis to the user’s needs [110,141]. All these factors considered, the software choice for nanopore sequence-interpretation will need to consider the nucleic acid-type of the sample, e.g., RNA, DNA, and methylation status, the speed capabilities of each computational pipeline, and the available computational resources available to the user. The latter two will heavily be determined by the sequencing and laboratory environment of the user.

Similarly, the decision to implement nanopore sequencing in research projects will be extremely user specific. Short read platforms benefit from their higher accuracy, though complicate long read assembly and the detection of structural variants and highly repetitive sequences. While ONT and PacBio both offer commercialized nanopore sequencing techniques and are improving error rates through platform and software improvements, they are offered with different cost and portability options. All of the above should be considered in deciding between them.

Finally, while we introduced a handful of recent applications of this technology, the list is by no means complete. The ability to sequence continuous long read assemblies alongside shorter sequences truly offers the potential of closing gaps in our knowledge of human genetics, transcriptomics, epigenetics, and infectious disease. In effect, this sequencing technology has the potential to permit the biochemical reconstitution of human disease processes. The standardization of library preparation techniques and bioinformatics pipelines, combined in particular with improvements in basecalling methods, will be an essential part of making this technology widely used as a molecular diagnostic tool.

## Figures and Tables

**Figure 1 micromachines-14-00459-f001:**
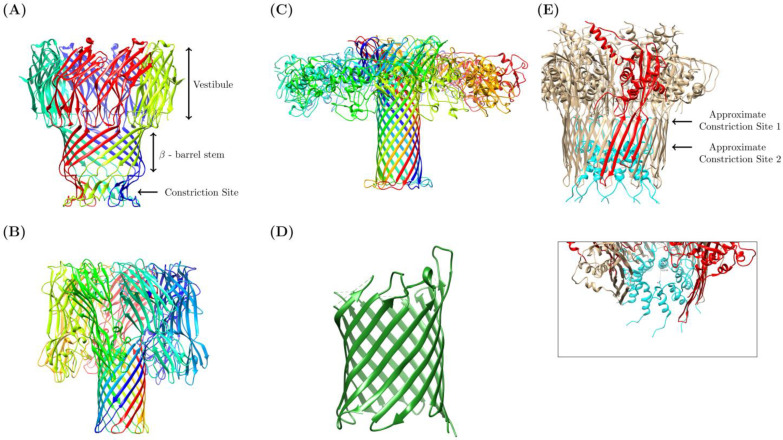
Anatomy of a Nanopore. A depiction of biological nanopores employed in sequencing. (**A**) MspA pore (PDB ID 1UUN) with a constriction site diameter of 1.2 nm, and a stem length of ~3.7 nm (**B**) Alpha Hemolysin (PDB ID 7AHL) with a constriction site diameter of 2.6 nm and a stem length of 5.2 nm (**C**) Aerolysin porin (PDB ID 5JZT) with a constriction site diameter of 1 nm, and a stem length ~10 nm. (**D**) A simplified depiction of OmpG pore (PDB ID 2F1C) with a constriction site of 1.3 nm. (**E**) CsgG-CsgF mutant (PDB ID 6SI7) with two constriction sites: the original CsgG constriction of 1 nm diameter (chain monomer of the original CsgG pore is depicted in red); and a secondary constriction caused by the insertion of CsgF (Cyan residues) with a 1.5 nm diameter. The insert shows the approximated diameter of the second constriction site, ~1.5 nm. Protein chains are depicted in different colors to help with distinction. All proteins were recreated in Chimera utilizing PDB IDs from published protein structures (noted PDB ID numbers).

**Figure 2 micromachines-14-00459-f002:**
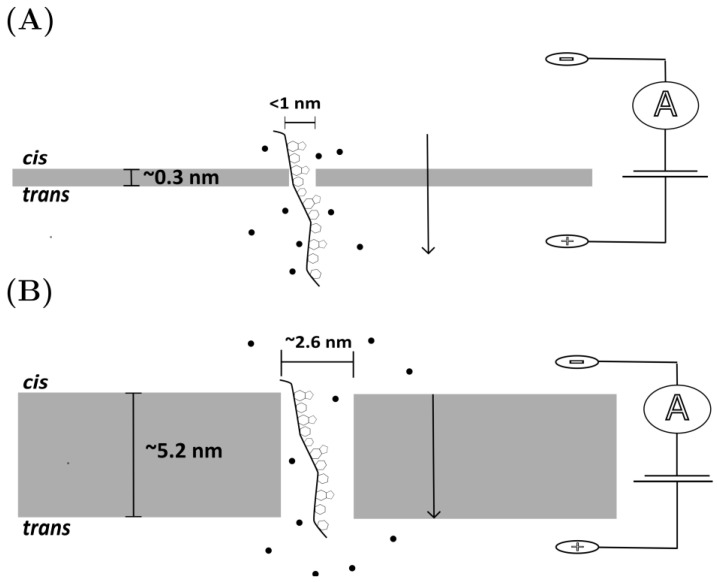
Visualizing Signal-to-Noise Ratio. The comparison of membrane thickness of (**A**) theoretical, ideal graphene monolayer dimensions (membrane thickness ~0.3 nm, with a pore diameter of <1 nmto (**B**) αHL pore dimensions (membrane thickness ~5.2 nm, pore diameter ~2.6 nm. Black arrows depict the direction of nucleic acid translocation from the cis to trans side of the membrane. In the case of the graphene monolayer membrane, only one to two bases contribute to the current disruption. Approximately 10 bases at one time can fit in the pore of αHL, all of them contributing to the signal. Figure 2 was generated in Inkscape 1.2 by the authors for demonstration purposes, and is not to scale.

## Data Availability

No new data were created or analyzed in this study. Data sharing is not applicable to this article.

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
