# Peer review of "An Introduction to Nanopore Sequencing: Past, Present, and Future Considerations"

_micromachines, 2023, doi:10.3390/mi14020459_

Round 1

Reviewer 1 Report

In the submitted manuscript, the authors provide an overview about the nanopore sequencing technology and its common applications specifically focusing on nucleic acid sequencing and  its clinical relevance. The authors describe the nanopore sequencing principles and the anatomy of nanopores and the materials used for its construction. The authors briefly discuss controlling sequencing noise and enhancing signal-to-noise ratio. The authors describe the different forms of library preparations used and their modifications. The authors discuss the clinical application of nucleic acid sequencing on nanopores such as fusion gene identification, short tandem repeat sequencing, epigenetic modifications and characterization  of virus and pathogen genomes. 

My major concern is  about the novelty of the review. 

In recent years, there have been numerous reviews about nanopore sequencing and its applications. I  found that the contents of this manuscript has already been covered in the previous published reviews such as below:

 https://www.nature.com/articles/s41587-021-01108-x

https://www.nature.com/articles/s41592-022-01633-w

https://www.nature.com/articles/s41565-022-01193-2

https://www.frontiersin.org/articles/10.3389/fgene.2022.1037134/full

The authors mention that the focus is mainly on the nucleic acid sequencing and its application. But, the majority of text is about DNA sequencing with very little or no mention to RNA sequencing. 

Minor concerns:

  1. The text size is not uniform throughout the article. 

Missing references for examples:

HomoPolish (line no. 526)

Proovread (line no. 536)

  1.  Also some of the acronyms are missing for example:

  2.  Short tandem repeat (line no. 443))

Author Response

We thank the reviewer for their time in raising the concerns about the novelty of the manuscript. We have a few comments regarding this issue and have made changes that may improve the comprehensibility of our review.

Comment:

My major concern is  about the novelty of the review. In recent years, there have been numerous reviews about nanopore sequencing and its applications. I  found that the contents of this manuscript has already been covered in the previous published reviews such as below: 

 https://www.nature.com/articles/s41587-021-01108-x

https://www.nature.com/articles/s41592-022-01633-w

https://www.nature.com/articles/s41565-022-01193-2

https://www.frontiersin.org/articles/10.3389/fgene.2022.1037134/full

 Response:

As the field of nanopore sequencing accelerates, largely motivated by Oxford Nanopore Technologies, more laboratories and investigators from a diverse array of research disciplines and backgrounds are beginning to consider it for their sequencing needs. Thus, the aim of our review was to provide a full introduction to the principles of nanopore sequencing for those with limited experience in the field.

The first review is an excellent one that we cite in our own review, which covers some of the biological nanopore developments and applications. However, it focuses largely on bioinformatics and data analysis techniques for ONT-specific applications and omits details on solid state nanopores, which is a field that has been gaining interest and investment by researchers and ONT themselves. What we have aimed to deliver in our own review is a broader perspective of the technology, discussing some of the history of protein and solid-state nanopores, library preparation basics, and clinical applications in addition to some commentary to the bioinformatics aspects.

The second review cited focuses exclusively on the direct RNA sequencing applications employed by Oxford Nanopore Technologies. While these are undeniably important factors for the experienced ONT or nanopore user to consider and it is an invaluable review, the goal of our paper is to introduce nanopore sequencing from multiple perspectives and applications, and combine these considerations in one review.

The third review focuses on areas that are not covered by our review. This paper delivers a comprehensive review of nanopore applications beyond DNA and RNA sequencing applications. However, our review puts a specific focus on the application of the technology for DNA and RNA sequencing, particularly in the realm of discovery and clinical applications such as biomarker detection and disease monitoring. In this way, the motivation of our review is to discuss the applications that this review purposefully excludes.

Response to reviewer 1 (cont.)

The fourth review identified reviews methods of modification assessment of nucleic acids, putting a strong emphasis on basecalling and bioinformatics tools. This is an essential part of nanopore sequencing and is important for readers to consider. However, our review serves more as a primer for nanopore sequencing as a whole, introducing and discussing each concept but not delving completely into any specific one.

To address the concerns of the reviewer in more detail, we would like to emphasize that our review brings in multiple concepts separately reviewed by others – protein pores, solid state pores, fabrication techniques for SSNPs, translocation control, library preparation considerations, basecalling consideration, and clinical applications – and combines these considerations into a single review. We believe that this marks our review apart from many others.

To aid in making our review more comprehensive in response to reviewer 1’s concerns, in this most recent revision, we have added a section comparing nanopore sequencing on ONT’s platforms to Illumina short-read and PacBio long-read sequencing (Section 5., lines 379 -501), two of the top competitors at the moment. We have inserted this section as a transition between our technical review of the sequencing platform, and our clinical applications section, which we believe improves the organization and flow of our review. We hope that this also addresses reviewer 1’s concerns regarding the organization and structure of the review, as well. This discussion has also been incorporated into the conclusion (lines 915-922).

Comment:

The authors mention that the focus is mainly on the nucleic acid sequencing and its application. But, the majority of text is about DNA sequencing with very little or no mention to RNA sequencing. 

Response: We thank the reviewer for this comment, and agree that our manuscript was criminally lacking in RNA sequencing applications. We have added examples of applications in full-transcript assembly (lines 787-799), microRNA detection (lines 809-829) and circRNA detection (lines 830-836) to try and balance our DNA and RNA sequencing sections. We have also expanded the emphasis we place on ONT’s direct RNA sequencing to underline this important application for readers (lines 767-785). Finally, we have included a brief example of single cell RNA sequencing on nanopore platforms (lines 838-848).

Minor concerns:

Comment:

The text size is not uniform throughout the article. 

Response: We thank the reviewer for noticing this discrepancy. We have gone through and reformatted the manuscript to be uniform in size and font and we improved our subject headings to clarify content and help with the organization of the review. Newly added or clarified subheadings can be found in lines 379, 392, 433, 503, 575, 612, 629, 657, 691, 742, 767, 787,

Response to Reviewer 1 (cont.)

809, 830, and 838. We hope these clearer separations will improve the organization of our manuscript.

Comment:

Missing references for examples:

HomoPolish (line no. 526)

Proovread (line no. 536)

Response: We thank the reviewer for catching this error. We have made corrections by adding these references to our text (line 762, line 803).

Comment: Also some of the acronyms are missing for example: 

Short tandem repeat (line no. 443))

Response: We thank the reviewer for catching this error. We have added an abbreviation for short tandem repeats at the first time it is employed in the text (line 659).

Reviewer 2 Report

Very well written and comprehensive review that covers the technology developments as well as the applications equally well. I think it is a good contribution to the literally and there is a shortage of good reviews of nanopore sequencing so this is welcome. I have a few comments on things that could be corrected/improved.

Line 347: PCR doesn't help avoid sample lost though the library preparation process. It's useful for low-input samples (whole-genome amplification) or low-abundance samples (target enrichment). Rare species could be interpreted to mean low frequency mutations for which PCR amplification would impede detection by increasing noise.

Line 407: Missing the obvious advancement of ultra-long library preparation for reads up to 1Mbp from Jain et al.

Line 346: This yield I think was only ever achieved once and ~1Gb much more common.

Line 510: Missing all the foundational work on Ebola/Zika virus on nanopore for real-time genomic surveillance which let into the SARS-CoV-2 and Monkeypox sequencing.

Line 518: MinION

Line 519: The virus is called SARS-CoV-2 the disease COVID-19.

Author Response

Response to Reviewer 2

We thank the reviewer for their time and attention in catching areas in our review that were lacking valuable information and references. We have made the following changes in accordance with their suggestions:

Comment:

Line 347: PCR doesn't help avoid sample lost though the library preparation process. It's useful for low-input samples (whole-genome amplification) or low-abundance samples (target enrichment). Rare species could be interpreted to mean low frequency mutations for which PCR amplification would impede detection by increasing noise.

Response:

We thank the reviewer for highlighting this discrepancy, and have clarified this sentence to read, “However, in the case of limited DNA input (<100 ng [81]), amplification is useful to provide acceptable library depth for whole genome or targeted sequencing applications.” (lines 541-542).

Comment:

Line 407: Missing the obvious advancement of ultra-long library preparation for reads up to 1Mbp from Jain et al.

Response:

We thank the reviewer for raising this concern, and we have clarified the development of the ultra-long read platform mentioned on lines 613-627, noting their development of the protocol, defining ultra-long reads to emphasize the difference between long and ultra-long read sequencing, (line 625) and clarifying their accomplishment in genome assembly (lines 625-627). We hope this fully answers the reviewer’s concerns for this reference.

Comment:

Line 346: This yield I think was only ever achieved once and ~1Gb much more common.

Response:

We thank the reviewer for pointing out this potentially misleading piece of information. We have remedied this by including a reference to the more common yield of 1 Gb, and have edited our language to say, “up to ~2.8 Gb of data (with yields commonly falling near 1 Gb [97])” (lines 648-649).

Response to Reviewer 2 (cont.)

Comment:

Line 510: Missing all the foundational work on Ebola/Zika virus on nanopore for real-time genomic surveillance which let into the SARS-CoV-2 and Monkeypox sequencing.

We thank the reviewer for their suggestions, and we have added references to the monitoring of Ebola, Influenza, and Zika viruses with MinION platforms to better contextualize the applications to SARS-CoV-2 and Monkeypox (lines 751-753).

Comment:

Line 518: MinION

Response:

We thank the reviewer for catching this typo. We have made the corrections to this (line 751).

Comment:

Line 519: The virus is called SARS-CoV-2 the disease COVID-19.

Response:

We thank the reviewer for catching this typo and have made this correction (line 754).

Author Response

Response to Reviewer 3

We thank the reviewer for their insight and suggestions to improve the comprehensibility of our review. We hope our revisions address their concerns and improve the utility of the manuscript for potential users of nanopore platforms.

Comment:

The manuscript describes full aspects of nanopore sequencing, but there are multiple third generation sequencing technologies not covered in Introduction section, such as PacBio, which is highly used in this area. The difference between these technologies and criteria to select Nanopore sequencing need to be proposed to facilitate readers to have a good option.

Response:

We thank the reviewer for this helpful insight into our approach of writing this review. We have inserted a new section (section 5) to discuss a brief comparison between specifically PacBio and ONT based sequencers, given their prominence in the field and similarity in performance. We provide some guidance for readers into where these platforms differ, which will inform their choice (section 5.2, lines 433-501). We have also included this in our conclusion (lines 915-922).

Comment:

Furthermore, as a review work, the comparison of short-read technologies with long-read nanopore sequencing is also need to be illustrated.

Response:

We thank the reviewer for this helpful suggestion and believe that the inclusion of our revision with regards to it will be helpful to readers. We have inserted an introductory comparison between long and short read platforms for the convenience of the novice reader (section 5-5.1., lines 379-431). This is also in our conclusion, (lines 904-911).

Comment:

Another recent single-cell sequencing technologies released by nanopore technology is missing, which has no requirement for fragmentation, nor read length limitations, sequences the entire RNA (cDNA) molecule

Response:

We thank the reviewer for their observation and have inserted a brief discussion of this landmark work in our RNA sequencing section (lines 838-848).

Response to Reviewer 3 (cont.)

Comment:

Figure 1 and 2 are not clear as to resolution and font size.

Response:

We thank the reviewer for their close attention to detail. We have resized and modified aspects of our figures through LaTex to be clearer in scaling and font sizes. Additionally, the captions have been resized to fit within the text, improving visibility. Finally, we have noted in the caption that these figures were not designed to be realistic in scale, but are generated for demonstration purposes (lines 300-306).

Comment:

In clinical application sections, the theory and procedure of basecalling and modification calling was thoroughly discussed. However, there are performance concerns for users about these areas, to my knowledge, e.g., Wick, Ryan R., (Performance of neural network basecalling tools for Oxford Nanopore sequencing. Genome biology 2019), and Liu, Yang, et al. (DNA methylation-calling tools for Oxford Nanopore sequencing: a survey and human epigenome-wide evaluation. Genome biology 2021). The criteria to choose which tools needs to be included.

Response:

We thank the reviewer for this very insightful comment into our discussion of basecalling methods for ONT. We have added a short comment in section 7.2 (lines 732-737) regarding these considerations raised by Wick and Liu and included a discussion in our conclusions section (lines 881-914), and hope that this will clarify for the reader the issues that often accompany basecalling technology currently available for use.

Comment:

In addition, the citation is not in a good style. It is close but not completely correct. It is noted that your manuscript needs careful editing by someone with expertise in technical English editing paying particular attention to English grammar, spelling, and sentence structure so that the goals and results of the study are clear to the reader.

Response:

We thank the reviewer for catching this error. We have gone through our citations and corrected them where necessary (notably including the correct citation for UCSF chimera, (citation 9, lines 85, 968-970). Additionally, we have systematically edited the spelling and grammar of the article, and have made changes where necessary including in:

Line 3: Capitalization: “Past, Present, and Future Considerations”

Response to Reviewer 3 (cont.)

Line 10: Improving language and sentence structure: “There has been significant progress made” instead of “There have been several developments”

Line 62: Added context to aid in content flow and connection to previous sections: “viral genome assemblies”

Line 158: word choice for clarification: “a pore produced by Mycobacterium smegmatis” to clarify the pore is produced by the organism.

Lines 191-192: Sentence structure for clarification: “If the translocation speed…” replaced the sentence fragment, “Too fast a translocation speed…”

Line 215: Grammar: “a” instead of “an”

Lines 287-288: sentence structure for clarification and content flow: Removing nanopores from the following sentence “materials of interest are silicon-based nanopores, glass nanopore capillaries…” to improve communication and limit repetition.

Line 310: Word choice for clarification: “thin dimensions” instead of “thin sizes”

Line 319: Word choice for clarification: “diameters of” instead of “within”

Line 329: Word choice for clarification: “e.g. pH and Temperature”

Line 333: Inclusion of “nanopores around 3.3 nm in size” instead of “nanopores ~3.3 nm” for sentence structure.

Line 373: “RNA” instead of “their” for clarity.

Line 522: Citations were corrected to include reference 82, and “review” changed to “reviews” to account for this.

Line 531: Citation correction.

Lines 608-609: Transition sentence added to clarify following content: “The applications mentioned refer to nanopore sequencing on ONT platforms, unless otherwise stated.”

Line 603: “Different” removed from “numerous different” to avoid redundancy.

Line 606: Run-on sentence was divided in two to help with content flow.

Line 640: Grammar: “a diagnostic”, not “diagnostic”

Line 654: “genomic and transcriptomic” instead of “genomics and transcriptomic”

Line 668: “nucleic acid bases” replaced “DNA bases” to clarify our meaning.

Line 684: “(discussed further below)” added to help connect content.

Response to Reviewer 3 (cont.)

Line 706: “MoS2” formatting corrected from “MoS2”

Line 742: “Infectious Disease Detection” instead of “Infectious Disease Characterization” altered to improve communication

Line 767: “RNA Sequencing” instead of “RNA Seq” to align with non-colloquial word use

Line 821: Sentence added to help with the content flow and clarify our meaning: “demonstrated that with library preparation modifications”

Line 924: “ability to” was added to improve sentence structure.

Line 925-926: “shorter sequences” instead of “shorter transcripts” to clarify that this applies to DNA as well as RNA.

Line 947: Contributions edited to represent editing efforts during revision.

Round 2

Reviewer 1 Report

Dear Authors, thank you for making the revisions. The  goal of the manuscript  is well defined in the revised edition. I believe that the manuscript will provide a comprehensive introduction to the readers new to the field of Nanopore. 

Reviewer 3 Report

The authors have been completely addressed by concerns. I can accept this version of revision.